# Geographical Disparities in Esophageal Cancer Incidence and Mortality in the United States

**DOI:** 10.3390/healthcare11050685

**Published:** 2023-02-25

**Authors:** Yeshwanth Vedire, Navpreet Rana, Adrienne Groman, Beas Siromoni, Sai Yendamuri, Sarbajit Mukherjee

**Affiliations:** 1Department of Thoracic Surgery, Roswell Park Comprehensive Cancer Center, Buffalo, NY 14263, USA; 2Department of Medicine, University at Buffalo School of Medicine, Buffalo, NY 14263, USA; 3Department of Biostatistics and Bioinformatics, Roswell Park Comprehensive Cancer Center, Buffalo, NY 14263, USA; 4School of Health Sciences, University of South Dakota, Vermillion, SD 57069, USA; 5Department of Hematology and Oncology, Roswell Park Comprehensive Cancer Center, Buffalo, NY 14263, USA

**Keywords:** esophageal cancer, geographic disparities, incidence trends, mortality trends

## Abstract

Background: Our previous research on neuroendocrine and gastric cancers has shown that patients living in rural areas have worse outcomes than urban patients. This study aimed to investigate the geographic and sociodemographic disparities in esophageal cancer patients. Methods: We conducted a retrospective study on esophageal cancer patients between 1975 and 2016 using the Surveillance, Epidemiology, and End Results database. Both univariate and multivariable analyses were performed to evaluate overall survival (OS) and disease-specific survival (DSS) between patients residing in rural (RA) and urban (MA) areas. Further, we used the National Cancer Database to understand differences in various quality of care metrics based on residence. Results: N = 49,421 (RA [12%]; MA [88%]). The incidence and mortality rates were consistently higher during the study period in RA. Patients living in RA were more commonly males (*p* < 0.001), Caucasian (*p* < 0.001), and had adenocarcinoma (*p* < 0.001). Multivariable analysis showed that RA had worse OS (HR = 1.08; *p* < 0.01) and DSS (HR = 1.07; *p* < 0.01). Quality of care was similar, except RA patients were more likely to be treated at a community hospital (*p* < 0.001). Conclusions: Our study identified geographic disparities in esophageal cancer incidence and outcomes despite the similar quality of care. Future research is needed to understand and attenuate such disparities.

## 1. Introduction

Esophageal cancer is the 8th most common cancer globally, with an age-standardized incidence rate (ASR) of 6.3 per 100,000 persons in 2020 [1]. As of 2022, the lifetime risk of developing esophageal cancer is 1 in 125 men and 1 in 417 women for the US population [2]. While the incidence and mortality trends of esophageal cancer in the US are decreasing, the global trends are reportedly increasing [3]. Age, gender, race, socioeconomic status, and geographical location have been reported to play a role in esophageal cancer incidence and mortality [3].

Males, Blacks people, people of lower socioeconomic status, and patients in low-income areas have been reported to be at a higher risk of developing and dying from esophageal cancer [3,4,5]. In contrast, a study in Brazil found an inverse relationship between esophageal cancer incidence and the level of urbanization [6]. A similar study utilizing the North American Association of Central Cancer Registries found no significant difference between overall cancer incidence rates between urban and rural areas. However, esophageal cancer incidence rates were higher in rural areas in the US [7].

A possible explanation for these disparities may be the difference in the quality of care. It has been well documented that Black patients are more likely to be diagnosed at a later stage and not receive timely definitive treatment resulting in poorer survival compared to Asian and White patients [8,9,10]. Other patient factors such as socioeconomic status, insurance status, and distance required to travel for medical care can influence the quality-of-care [11,12]. Interestingly, Clark et al. found that patients at high-volume academic centers had better outcomes than low-volume community centers [12].

Our group has previously used the Surveillance Epidemiology and End Results (SEER) and the National Cancer Database (NCDB) databases to explore trends and disparities in neuroendocrine [13] and gastric cancers [14] between urban and rural populations in the US. We sought to assess if any such disparities exist for esophageal cancer by analyzing data from the SEER and NCDB databases.

## 2. Materials and Methods

### 2.1. Data Source

The data for this retrospective analysis were extracted from the Surveillance, Epidemiology, and End Results Program (SEER) database from 1975 to 2016 and National Cancer Data Base (NCDB) from 2006 to 2017. The SEER database is a National Cancer Institute program that collects cancer-related data from various population-based registries, which cover approximately 47.9% of the US population [15]. The SEER database collects patient demographics, primary tumor site, tumor morphology, stage at diagnosis, course of treatment, insurance status, patient location, vital status, and survival data. The data on cancer rates and mortality are received from the Census Bureau and Nations Center for Health Statistics.

The NCDB is a joint effort by the American College of Surgeons and the American Cancer Society to collect data from hospital cancer registries to evaluate cancer trends and treatment patterns [16]. The NCDB captures data from approximately 1500 commission-on cancer-accredited facilities covering nearly 70% of newly diagnosed cancer patients.

### 2.2. Study Population

We used the International Classification of Diseases of Oncology, 3rd Edition (ICD-O-3) diagnostic codes to identify and include all esophageal cancer patients from NCDB and SEER databases for our analysis. Patients from all stages (AJCC 6th and 7th editions) were included in the analysis.

The residential area of patients was classified as urban or rural based on the Rural-Urban Continuum Code available (RUCC) in the NCDB and SEER databases. The RUCC codes were used to categorize geographical localities into metropolitan and non-metropolitan by the Office of Management and Budget based on population. Consistent with our previous research, we categorized counties as urban if they were considered metropolitan (MA) as per RUCC coding (RUCC 1–3) and counties as rural if they were considered non-metropolitan (RA) as per RUCC coding (RUCC 4–9) [14].

### 2.3. Statistical Analysis

The SEER database was utilized to identify and analyze data on patient demographics such as age, race, sex, insurance (insured, uninsured, and unknown), residence (metro [MA], and rural [RA]), marital status, tumor characteristics (histology, grade, and stage), period of diagnosis (1975–1989, 1990–2000, 2001–2010, and 2011–1016) patient vital status, and disease-specific (DSS) overall survival (OS). The incidence and mortality rates from the various time periods were calculated to analyze esophageal cancer trends and evaluate the difference in trends between the rates and survival outcomes in RA and MA populations.

Similar sociodemographic data were collected for patients in the NCDB database, which included patient age, ethnicity (Hispanic and non-Hispanic), race, sex, insurance provider (government, private, and uninsured), county median income (≤USD 50,353 and ≥USD 50,354), residence (MA, and RA), facility at which treated (academic/Integrated, Community, and unknown), distance traveled for care (miles), tumor characteristics (histology, grade, and stage), period of diagnosis (2006–2011 and 2012–2017), OS data, and quality of care indicators such as the number of regional lymph nodes examined (<15 and ≥15), time from diagnosis to start of treatment, adjuvant and neoadjuvant therapy received (yes, and no), chemotherapy received (none, single agent, multiagent, unknown regimen, unknown if chemotherapy was received), surgical margins checked (yes or no), length of inpatient stay, 30-day readmission (planned and unplanned), and 30- and 90-day mortality.

Association between the place of residence and various sociodemographic variables, tumor characteristics, and quality of care metrics were assessed using Wilcoxon Rank Sum (continuous variable) and Chi-square tests (categorical variables). The study’s primary goals were to evaluate incidence and mortality trends in the rural and urban population between 1975 and 2016 and to estimate the OS and DSS using univariate and multivariate Cox proportional modeling. The multivariate model adjusted survival for age, sex, stage, grade, year of diagnosis, insurance status, marital status, race, and area of residence. Using the log-rank test, Kaplan Meir survival analysis was used to compare long-term outcomes between urban and rural areas. Incidence rates were calculated for each residence (MA and RA) and decade using the SEER population database. Data regarding RUCC codes were available for 676 and 2718 cases in the SEER and NCDB databases, respectively. These cases were excluded from all analyses. Statistical significance was indicated by *p* < 0.05. All statistical analyses were performed using SAS, version 9.4, statistical software (SAS Institute Inc., Cary, NC, USA).

### 2.4. Reporting Guidelines

This study is reported as per Strengthening the Reporting of Observational Studies in Epidemiology (STROBE) guidelines for cohort studies (Appendix A).

## 3. Results

### 3.1. SEER Database

A total of 49,421 esophageal cancer patients with RUCC codes were identified in our retrospective analysis of the SEER database between 1975 and 2016. The mean age of the cohort was 65.4 years. Most of the patients were males (78.6%), Caucasian (75%), and had an urban residence (87.5%). A total of 44,048 (87.9%) of the patients died in 41 years follow-up period.

Descriptive characteristics of patients residing in an MA (87.5%) and RA (12.5%) are compared and summarized in Table 1. Patients in RA were more likely to be males (RA vs. MA, 82.1% vs. 78.1%; Chi-Square test, *p* < 0.001), Caucasian (86.4% vs. 74.2%; *p* < 0.001), married (60.5% vs. 55.4%; *p* < 0.001), and have adenocarcinoma (64.2% vs. 56.9%; *p* < 0.001). Although there was a statistically significant difference in patient insurance status and tumor grade at diagnosis between people residing in RA and MA, data were not known for a significant portion of the population for both characteristics. There was no significant difference in patient age, tumor stage, and the number of patient deaths.

Chi Square and Wilcoxon Rank Sum tests were performed to compare sociodemographic and clinicopathological variables between urban and rural esophageal cancer patients. All significantly different (*p* < 0.05) are highlighted in bold.

Esophageal cancer patients residing in an MA had consistently lower age-adjusted incidence rates between 1975 and 2016 than patients with rural residences. The incidence rates in patients from an RA showed an upward trend with a rate of 4.66 cases/100,000 people between 1975 and 1989 to 6.40 cases/100,000 people between 2011 and 2016, whereas the rate in MA was relatively stable with 2.39 cases/100,000 people between 1975 and 1989 to 3.07 cases/100,000 people between 2011 and 2016. Similar to incidence rates, age-adjusted mortality rates were also consistently higher in RA patients. However, unlike incidence rates, mortality rates were relatively stable in both RA and MA patients. Incidence and mortality rates in RA and MA populations are shown in Figure 1 and Appendix A.

In addition to comparing the trends between rural and urban populations, we performed attributable risk percentage and population attributable risk percent calculations between these two populations. The attributable risk percentage and the population attributable risk percent for esophageal cancer incidence ranged from 30.20 to 61.90 and from 1.39 to 6.98 between 1975 and 2016, respectively. The table with attributable risk percentage and population attributable risk for every year between 1975 and 2016 is presented in Appendix A.

We performed univariate (Table 2) and multivariable survival analyses (Table 3) for OS and DSS for esophageal cancer patients. On univariate analysis for OS, increasing age (HR [95% CI], 1.01 [1.01–1.01]; Wald *p* < 0.001), African American race (1.37 [1.33–1.40]; *p* < 0.001), single (1.27 [1.25–1.30]; *p* < 0.001), and uninsured (1.43 [1.32–1.54]; *p* < 0.001) patients were associated with poor outcomes. In addition, tumors with squamous cell carcinoma histology (1.30 [1.28–1.33]; *p* < 0.001), grade III/IV (1.29 [1.27–1.32]; *p* < 0.001), and regional (1.32 [1.29–1.36]; *p* < 0.001) and distant (2.76 [2.70–2.83]; *p* < 0.001) spread were also associated with poorer outcomes. Patient sex and location of residence were not significant predictors of OS. Similar to OS, worse DSS was associated with patient age (1.01 [1.01–1.01]; *p* < 0.001), African American race (1.37 [1.33–1.41]; *p* < 0.001), single marital status (1.26 [1.23–1.28]; *p* < 0.001), uninsured status (1.44 [1.32–1.56]; *p* < 0.001), and tumors with squamous cell carcinoma histology (1.28 [1.25–1.31]; *p* < 0.001). Higher grade (II/IV) (1.35 [1.32–1.38]; *p* < 0.001), regional (1.48 [1.44–1.52]; *p* < 0.001), and distant (3.28 [3.19–3.37]; *p* < 0.001) stages were found to be poor indicators on univariate analysis (Table 3). The location of the residence was not associated with either OS or DSS (Appendix A).

Univariate cox proportional modeling for OS and DSS with HR and 95% CI for sociodemographic and clinicopathological variables available in the SEER database are shown. All statistically significant (*p* < 0.05) outcomes are highlighted in bold.

For multivariable analysis, age, sex, race, marital status, insurance status, tumor histology, grade and stage, residence, and year of diagnosis were used as covariates. Multivariable analysis confirmed the results of univariate analysis for age, race, marital and insurance status, tumor histology, stage and grade, and year of diagnosis as significant prognostic indicators for both OS and DSS. Additionally, the female sex was found to be associated with better OS (0.87 [0.85–0.90]; *p* < 0.001) and DSS (0.90 [0.87–0.92]; *p* < 0.001). In contrast to univariate analysis, patients residing in RA had a significantly poorer OS (1.07 [1.04–1.10]; *p* < 0.001) and DSS (1.08 [1.04–1.11]; *p* < 0.001) on multivariable analysis (Table 4; Figure 2).

Multivariate cox proportional modeling for OS and DSS with HR and 95% CI for sociodemographic and clinicopathological variables available in the SEER database are shown. All variables collected from the SEER database were used as covariates in multivariate model. All statistically significant outcomes (*p* < 0.05) are highlighted in bold.

### 3.2. NCDB

To better understand the difference in incidence and mortality rates and survival analyses observed in the SEER data between patients residing in RA and MA, we analyzed the quality-of-care variables available in the NCDB database to try and explain these differences. A total of 72,226 esophageal cancer patients with RUCC codes were identified in our retrospective analysis; 12,930 (17.9%) had a rural residence; and 59,296 (82.1%) of the patients resided in an urban area. Data about treatment facility, time from diagnosis to treatment, type of chemotherapy, sequence of radiation therapy, time from diagnosis to surgery, number of lymph nodes examined, surgical margin status, length of stay, planned or unplanned 30-day readmission, and 30- and 90-day mortality were evaluated as a measure of the quality of care from NCDB for patients diagnosed between 2006 and 2017.

We saw a statistically significant difference between RA and MA patients for most of the quality-of-care variables, as shown in Table 4. However, a clinically significant difference was found only for county median income, type of insurance, distance traveled for treatment, and type of treatment facility. Patients in RA were more likely to live in counties with a median income ≤ USD 50,353 (72.5% vs. 36.9%; *p* < 0.001) and were insured by a government entity (61.6% vs. 57.2%; *p* < 0.001). Furthermore, they traveled further to receive care (Mean in miles [Std. Deviation], 67.1 [130.1] vs. 27.0 [107.4]; Wilcoxon rank Sum test *p* < 0.001) and received care at a community facility (57.3% vs. 43.9%; *p* < 0.001).

Chi Square and Wilcoxon Rank Sum tests were performed to compare quality of care variables between urban and rural esophageal cancer patients. All significantly different (*p* < 0.05) are highlighted in bold.

## 4. Discussion

In this population-based retrospective analysis of the SEER and NCDB databases, we found that esophageal cancer incidence and mortality rates steadily increased from 1975 to 2016 in both rural and urban areas. Over this period, patients residing in RA consistently had higher incidence and mortality rates. Interestingly, DSS and OS were not associated with residence on univariate analysis. However, on multivariable analysis for DSS and OS, RA patients had an HR of 1.08 (1.04–1.11) and 1.07 (1.04–1.10), respectively. This suggested that other variables and factors may contribute to the differences in survival. To possibly explore these factors, we analyzed differences in variables that reflected the quality of care between RA and MA patients and found that RA patients received a similar quality and type of treatment as MA patients. This suggests that a combination of factors may explain these discrepancies.

Our study shows that the age adjusted incidence and mortality rates in both urban and rural populations increased consistently between 1975 and 2016; this is in contrast to the study performed by Ulhenhopp et al. This study shows a downward trend for both the incidence and mortality rates between a similar time period using the SEER database. A possible explanation for this could be that our study included patients in SEER and NCDB for whom RUCC codes were available.

We found that patients residing in RA were more likely to be males. Sociodemographic factors have been reported to play a significant role in esophageal cancer incidence, treatment, and survival [17]. Studies have reported a male-to-female incidence ratio of 9:1 for esophageal adenocarcinoma [18,19] and a higher incidence of high-grade disease in males [20]. Differences in hormonal levels of estrogen and insulin, growth factors such as IGF-1, and inflammatory mediators have been proposed as possible explanations for these differences in esophageal and other cancer incidence and survival [18,21].

A study describing costs of care at various stages of treatment for different cancers reported that an initial and end-of-life care in esophageal cancer patients was USD 20,433 and USD 18,760, respectively, one of the highest across various cancers [22]. A study examining colorectal, lung, cervical, and breast cancer trends in the US found that uninsured patients with decreased or no physician contact were less likely to undergo age-appropriate screening for cancer [23]. While insured patients showed better outcomes than uninsured patients, insurance type is also a significant predictor of survival [24]. In our analysis, we found that RA patients were more likely to have a lower income and more likely to be either uninsured or insured by a government agency which could explain the worse survival in the rural population.

Quality of care disparities can explain the differences observed across socioeconomic strata. Patients having a lower socioeconomic status are more likely to be victims of these disparities and have poorer outcomes [25]. These disparities may stem from decreased availability of high-quality care or increased difficulty accessing such care. In our study, we found that patients residing in RA were more likely to travel farther and receive care at a community center than their MA counterparts who received care at integrated academic institutions. Although RA patients were more likely to be treated at community centers, they had a similar 30-day unplanned readmission and 30-day mortality as urban patients, but 90-day mortality was higher. This observation was similar to the study reported by Boffa et al. They found that patients treated at affiliate hospitals had better surgical margins, a similar 30-day mortality rate, and a higher 90-day mortality rate [26]. These results are hypothesis generating that immediate peri-operative care always do not translate into long term outcomes in esophageal cancer. Another advantage commonly stated with surgical treatment at academic centers is the improved mortality rates with increased annual hospital and surgeon volumes, as seen in a meta-analysis by Brusselaers et al. [27].

The Leapfrog Group, an advocacy organization, suggested a minimum hospital volume and surgeon volume of 20 and seven, respectively, for esophagectomies [28]. While adopting such standards might not decrease the average cost of an esophagectomy, higher hospital and surgeon volumes have decreased complications and length of stay, which are the biggest drivers of cost [12,29,30]. Although no federal mandate exists in the US towards regionalization, there has been a 12.4% decline in the number of centers offering esophagectomy between 2004 and 2012 [31]. This consolidation of esophagectomy centers was associated with fewer patients treated at low-volume centers, improved 90-day mortality rate, lymph node harvest, and decreased length of stay and positive margin rate. While regionalization brings improved outcomes and decreased medical costs, robust structures and strategies must be implemented to decrease the risk of further marginalizing socioeconomically disadvantaged sections of society from accessing quality care.

To our knowledge this is the first population-based study investigating the disparities in incidence and mortality trends of esophageal cancer between RA and MA populations using national databases in the US. We also evaluated how sociodemographic variables impact patients’ overall and disease-specific survival. Additionally, we used the NCDB database to identify differences in quality-of-care metrics, which might explain the difference in survival observed between the two populations. However, our study has limitations, including missing and unknown data, most notably for the stage, grade, insurance status and treatment specifics, and positive margin rate. Secondly, selection and misclassification bias may have impacted the study, given its retrospective nature. Although we used previously reported definitions for rural and urban areas, the differing definitions of rurality may cause a misclassification bias [13,14].

## 5. Conclusions

Our SEER-based analysis found significant sociodemographic differences between esophageal cancer patients in RA vs. MA. We found that despite the advances in diagnostic and treatment techniques, the incidence and mortality rates increased between 1975 and 2016. Additionally, the rate of increase and the absolute rates were higher in RA consistently over this period. Multivariable survival analysis showed significantly poor overall and disease-specific survival in RA patients. Although quality of care metrics were similar between the two populations, a larger proportion of the population being males, lower median income, and socioeconomic status, difficulty accessing care, and treatment at community centers amongst rural patients could be some of the possible explanations for the observed disparities in incidence, mortality rates, and survival between the two populations in the US. Our results are consistent with similar studies in other countries and studies in the US evaluating other cancers [4,5,6,13,14]. Our findings suggest future research with more robust datasets are required to understand the underpinnings of the observed disparities. This understanding can be used to develop tailored healthcare policies needed to improve the quality of care for all esophageal cancer patients in the US.

## Figures and Tables

**Figure 1 healthcare-11-00685-f001:**
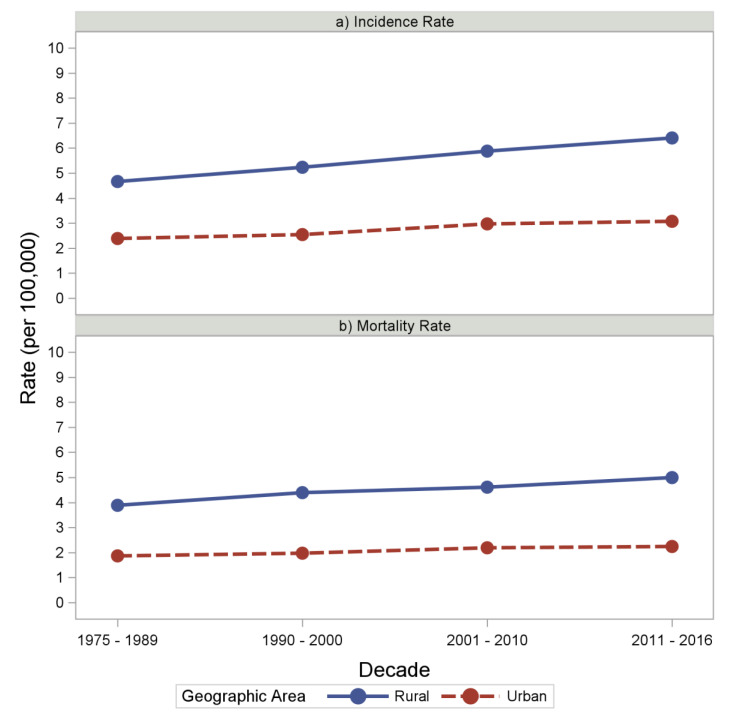
Incidence and Mortality trends of esophageal cancer between 1975 and 2016. (**a**) The trends of incidence rates per 100,000 population in rural and urban areas between 1975 and 2016. (**b**) The trends of mortality rates per 100,000 patients in rural and urban areas between 1975 and 2016.

**Figure 2 healthcare-11-00685-f002:**
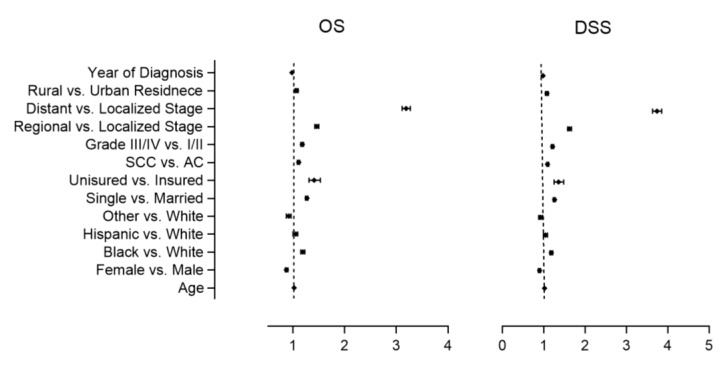
Multivariable OS and DSS forest plots with 95% CI for sociodemographic and clinicopathological variables available in the SEER database are shown. All variables collected from the SEER database were used as covariates in multivariate model.

**Table 1 healthcare-11-00685-t001:** SEER Database cohort descriptive characteristics.

Characteristics	All ^1^ (N = 49,421)	Rural ^1^ (N = 6199)	Urban ^1^ (N = 43,222)	*p* Value ^2^
Age	65.4 (11.5)	65.1 (11.2)	65.4 (11.5)	0.08
Sex				<0.001
Male	39,367 (78.6%)	5088 (82.1%)	33,737 (78.1%)	
Female	10,730 (21.4%)	1111 (17.9%)	9485 (21.9%)	
Race				<0.001
non-Hispanic White	37,564 (75%)	5356 (86.4%)	32,067 (74.2%)	
non-Hispanic Black	6751 (13.5%)	549 (8.9%)	6193 (14.3%)	
Hispanic	3078 (6.1%)	172 (2.8%)	2893 (6.7%)	
Other	2704 (5.4%)	122 (2%)	2069 (4.8%)	
Marital Status				<0.001
Single	22,037 (44%)	2448 (39.5%)	19,285 (44.6%)	
Married	28,060 (56%)	3751 (60.5%)	23,937 (55.4%)	
Insurance				<0.001
Uninsured	786 (1.6%)	142 (2.3%)	643 (1.5%)	
Insured	20,639 (41.2%)	2760 (44.5%)	17,836 (41.3%)	
Unknown	28,672 (57.2%)	3297 (53.2%)	24,743 (57.2%)	
Histology				<0.001
Adenocarcinoma	28,675 (57.2%)	3977 (64.2%)	24,584 (56.9%)	
Squamous Cell Carcinoma	21,422 (42.8%)	2222 (35.8%)	18,638 (43.1%)	
Grade				<0.001
I/II	19,950 (39.8%)	2535 (40.9%)	17,109 (39.6%)	
III/IV	21,052 (42%)	2710 (43.7%)	18,071 (41.8%)	
Unknown	9095 (18.2%)	954 (15.4%)	8042 (18.6%)	
Stage				0.1
Localized	13,436 (26.8%)	1710 (27.6%)	11,509 (26.6%)	
Regional	17,193 (34.3%)	2132 (34.4%)	14,840 (34.3%)	
Distant	19,468 (38.9%)	2357 (38%)	16,873 (39%)	
Dead				0.9
Yes	44,048 (87.9%)	5441 (87.8%)	37,956 (87.8%)	
No	6049 (12.1%)	758 (12.2%)	5266 (12.2%)	

^1^ Mean (Std. Deviation); n (%) ^2^ Chi Square test; Wilcoxon Rank Sum test.

**Table 2 healthcare-11-00685-t002:** Univariate cox proportional modeling for OS and DSS in the SEER cohort.

Variables	Overall Survival HR (95% CI)	*p* Value ^1^	Disease Specific Survival HR (95% CI)	*p* Value ^1^
Age	1.01	<0.001	1.01	<0.001
	(1.01–1.01)		(1.01–1.01)	
Sex				
Female vs. Male	1.01	0.3	1.01	0.4
	(0.99–1.04)		(0.99–1.04)	
Race				
non-Hispanic Black vs. non-Hispanic White	1.37	<0.001	1.37	<0.001
	(1.33–1.40)		(1.33–1.41)	
Hispanic vs. non-Hispanic White	1.04	0.05	1.04	0.07
	(1.00–1.08)		(1.00–1.08)	
Other vs. non-Hispanic White	0.97	0.2	0.98	0.4
	(0.93–1.02)		(0.93–1.03)	
Marital Status				
Single vs. Married	1.27	<0.001	1.26	<0.001
	(1.25–1.30)		(1.23–1.28)	
Insurance status				
Uninsured vs. Insured	1.43	<0.001	1.44	<0.001
	(1.32–1.54)		(1.32–1.56)	
Histology				
Squamous Cell Carcinoma vs. Adenocarcinoma	1.3	<0.001	1.28	<0.001
	(1.28–1.33)		(1.25–1.31)	
Grade				
III/IV vs. I/II	1.29	<0.001	1.35	<0.001
	(1.27–1.32)		(1.32–1.38)	
Stage				
Regional vs. Localized	1.32	<0.001	1.48	<0.001
	(1.29–1.36)		(1.44–1.52)	
Distant vs. Localized	2.76	<0.001	3.28	<0.001
	(2.70–2.83)		(3.19–3.37)	
Residence				
Rural vs. Urban	1.01	0.5	1.01	0.5
	(0.98–1.04)		(0.98–1.04)	
Year of diagnosis	1.01	<0.001	0.98	<0.001
	(1.01–1.01)		(0.98–0.98)	

^1^ Wald *p*.

**Table 3 healthcare-11-00685-t003:** Multivariate cox proportional modeling for OS and DSS in the SEER cohort.

Variables	Overall Survival HR (95% CI)	*p* Value ^1^	Disease Specific Survival HR (95% CI)	*p* Value ^1^
Age	1.02	<0.001	1.02	<0.001
	(1.02–1.02)		(1.02–1.02)	
Sex				
Female vs. Male	0.87	<0.001	0.9	<0.001
	(0.85–0.90)		(0.87–0.92)	
Race				
non-Hispanic Black vs. non-Hispanic White	1.19	<0.001	1.18	<0.001
	(1.15–1.23)		(1.15–1.22)	
Hispanic vs. non-Hispanic White	1.05	0.02	1.04	0.08
	(1.00–1.09)		(0.99–1.09)	
Other vs. non-Hispanic White	0.92	<0.001	0.92	0.003
	(0.87–0.96)		(0.88–0.97)	
Marital Status				
Single vs. Married	1.27	<0.001	1.26	<0.001
	(1.24–1.29)		(1.23–1.28)	
Insurance status				
Uninsured vs. Insured	1.41	<0.001	1.36	<0.001
	(1.31–1.53)		(1.25–1.48)	
Histology				
Squamous Cell Carcinoma vs. Adenocarcinoma	1.11	<0.001	1.09	<0.001
	(1.08–1.13)		(1.07–1.12)	
Grade				
III/IV vs. I/II	1.18	<0.001	1.21	<0.001
	(1.15–1.20)		(1.18–1.24)	
Stage				
Regional vs. Localized	1.46	<0.001	1.62	<0.001
	(1.42–1.50)		(1.58–1.67)	
Distant vs. Localized	3.19	<0.001	3.74	<0.001
	(3.11–3.27)		(3.63–3.85)	
Residence				
Rural vs. Urban	1.07	<0.001	1.08	<0.001
	(1.04–1.10)		(1.04–1.11)	
Year of diagnosis	0.98	<0.001	0.98	<0.001
	(0.98–0.98)		(0.98–0.98)	

^1^ Wald *p*.

**Table 4 healthcare-11-00685-t004:** Descriptive characteristics of the quality-of-care variables in the NCDB cohort.

Characteristics	All ^1^ (N = 72,226)	Rural ^1^ (N = 12,930)	Urban ^1^ (N = 59,296)	*p* Value ^2^
Median Income				<0.001
≤USD 50,353	30,103 (43.1%)	8677 (72.5%)	20,516 (36.9%)	
≥USD 50,354	39,760 (56.9%)	3296 (27.5%)	35,105 (63.1%)	
Insurance				<0.001
Uninsured	2675 (3.6%)	494 (3.8%)	2122 (3.6%)	
Private	27,154 (36.2%)	4158 (32.2%)	21,904 (36.9%)	
Government	43,199 (57.6%)	7961 (61.6%)	33,924 (57.2%)	
Unknown	1916 (2.6%)	317 (2.5%)	1346 (2.3%)	
Distance traveled for care (miles)	34.2 (112.9)	67.1 (130.1)	27.0 (107.4)	<0.001
Treatment facility				<0.001
Community	34,255 (45.7%)	7415 (57.3%)	26,042 (43.9%)	
Academic/Integrated	39,730 (53.0%)	5358 (41.4%)	32,503 (54.8%)	
Unknown	959 (1.3%)	157 (1.2%)	751 (1.3%)	
Time from diagnosis to treatment start (days)	36.5 (35.8)	36.8 (43.1)	36.5 (33.9)	0.1
Time from diagnosis to chemotherapy (days)	40.8 (36.2)	41.6 (47.0)	40.6 (33.3)	0.03
Type of chemotherapy				<0.001
No chemotherapy	20,486 (27.3%)	3293 (25.5%)	16,291 (27.5%)	
Single-agent chemotherapy	3872 (5.2%)	701 (5.4%)	3052 (5.1%)	
Multiagent chemotherapy	38,436 (51.3%)	7000 (54.1%)	30,229 (51.0%)	
Unknown Chemotherapy	4151 (5.5%)	589 (4.6%)	3265 (5.5%)	
Unknown if chemotherapy received	7999 (10.7%)	1347 (10.4%)	6459 (10.9%)	
Radiation sequence				<0.001
No radiation	59,793 (79.8%)	10,088 (78.0%)	47,606 (80.3%)	
Radiation before surgery	11,652 (15.5%)	2182 (16.9%)	8947 (15.1%)	
Radiation after surgery	2200 (2.9%)	428 (3.3%)	1703 (2.9%)	
Radiation before and after surgery	97 (0.1%)	19 (0.1%)	75 (0.1%)	
Intraoperative radiation	3 (<0.1%)	1 (<0.1%)	2 (<0.1%)	
Intraoperative radiation with other therapy	4 (<0.1%)	2 (<0.1%)	2 (<0.1%)	
Unknown sequence	1195 (1.6%)	210 (1.6%)	961 (1.6%)	
Time from diagnosis to surgery (days)	98.3 (68.7)	99.1 (65.3)	98.2 (69.4)	0.1
Number of regional lymph nodes examined				0.4
< 15	62,607 (83.5%)	10,809 (83.6%)	49,843 (84.1%)	
≥ 15	7770 (10.4%)	1340 (10.4%)	5998 (10.1%)	
Unknown	4567 (6.1%)	781 (6.0%)	3455 (5.8%)	
Positive surgical margins				<0.001
No	21,110 (28.2%)	3748 (29.0%)	16,281 (27.5%)	
Yes	1660 (2.2%)	304 (2.4%)	1294 (2.2%)	
Unknown	51,174 (69.6%)	8878 (68.7%)	41,721 (70.4%)	
Duration of inpatient hospital stay	11.2 (13.3)	11.7 (13.3)	11.1 (13.2)	<0.001
30 day readmission				0.2
Unplanned	1656 (2.2%)	266 (2.1%)	1310 (2.2%)	
Planned or not readmitted	73,288 (97.8%)	12,664 (97.9%)	57,986 (97.8%)	
30 day mortality				0.2
Alive	21,466 (89.6%)	3779 (89.2%)	16,536 (89.3%)	
Dead	662 (2.8%)	135 (3.2%)	506 (2.7%)	
No surgery or < 30 day follow up	1837 (7.7%)	321 (7.6%)	1465 (7.9%)	
90 day mortality				0.02
Alive	20,603 (86.0%)	3596 (84.9%)	15,888 (85.8%)	
Dead	1417 (5.9%)	294 (6.9%)	1082 (5.8%)	
No surgery or <90 day follow up	1945 (8.1%)	345 (8.1%)	1537 (8.3%)	

^1^ Mean (Std. Deviation); n (%) ^2^ Chi Square test; Wilcoxon Rank Sum test.

## Data Availability

All the data used for this study are from the publicly available SEER and NCDB databases.

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
