# Peer review of "Geographical Disparities in Esophageal Cancer Incidence and Mortality in the United States"

_healthcare, 2023, doi:10.3390/healthcare11050685_

Round 1

Reviewer 1 Report

Major points:

1. Please explain the time frame 1975-2016 chosen for analysis. What is about the last six years.

2. In the Discussion please explain in more detail the potential mechanism for the disparities of distribution, incidence and mortality of this cancer. 

Author Response

Please refer to attached word document.

Reviewer 2 Report

This is a retrospective study of the disparities in esophageal cancer between urban and rural residents. It was found that both the incidence and mortality were higher in rural than in urban of the United States despite the similar quality of care. The study is based on a population of 49421, with 12% living in rural areas and the remainder in the cities. First of all, it is hard to generate a meaningful conclusion when one group was 7 times bigger than the other. Secondly, the authors could not explain why there were such disparities. In the introduction section, the authors cited an article saying that both the incidence and the mortality of esophageal cancer in the USA were declining, but their study found both increasing. The authors did not explain this disparity. Moreover, this study did not generate any new meaningful information other than what is already known. 

Author Response

Please refer to attached word document.

Reviewer 3 Report

Thank you for the opportunity to review the manuscript. Overall, a current topic for a broader readership and further exploration of this topic is certainly unique, especially to investigate the geographic and sociodemographic disparities in esophageal cancer patients in the United States.

A few questions / comments and suggestions:

In Line 35-55, the theoretical Introduction is very brief and needs to be developed a bit further, relevant to the study is not clear.

In Line 260, what are the meanings of long-term outcomes in esophageal cancer, relevant to the study is not clear.

The paper at present is quite dense that is uniquely focused, and this would include a separate section on implications for this manuscript.

Author Response

Please refer to attached word document.

Reviewer 4 Report

Vedire et al submit a descriptive epidemiological study about esophageal cancer in the USA. Overall the study is well designed, statistically sound with proper methodology. There are some minor spellings and english language edits needed in the abstract and the introduction.

There are several comments that I can suggest to the authors in order to increase the merit of their manuscript.

1. The HR/multivariate analysis can be plotted as a forest plot with 95% CI in order to make the presentation and the attention of the reader greater.

2. The authors could also include some representative KM plots in the proper statistics, based on factors that stratify the survival of the patients.

3. The authors can also perform statistical analysis to association of specific demographic and social factors with more aggressive grade and stage, know determinants of poor outcomes in esophageal cancer. 

4. This study identifies several SES factors that negatively affect the outcome of patient with esophageal cancer. It would be very important and interesting for the authors to include calculations about the Attributed Percentage Risk and Population Attributed Risk % for some of these factors. This information will be a powerful public health tool to identify the effect of the social disparities in the US and will guide us for future public policies.

I would like to wish best of luck to the authors for their revision process.

Author Response

Please refer to attached word document.

Round 2

Reviewer 2 Report

The issues remain still.

Reviewer 4 Report

Vedire et al, resubmit a drastically improved version of their manuscript. The auhtors have revised all the topics that were suggested and included additional analysis as suggested during the first revision session (univariate-multivariate cox regression hazard, PAR%) . The new analysis has been done in a scientifically accurate manner and the new results are clearly interpreted in the text. At this point, the article is in an adequate state for publication.

I would like to congratulate the authors for their efforts.